# Daily Step Counts from the First Thailand National Steps Challenge in 2020: A Cross-Sectional Study

**DOI:** 10.3390/ijerph17228433

**Published:** 2020-11-14

**Authors:** Thitikorn Topothai, Rapeepong Suphanchaimat, Viroj Tangcharoensathien, Weerasak Putthasri, Thitiporn Sukaew, Udom Asawutmangkul, Chompoonut Topothai, Peeraya Piancharoen, Chonlaphan Piyathawornanan

**Affiliations:** 1International Health Policy Program, Ministry of Public Health, Nonthaburi 11000, Thailand; rapeepong@ihpp.thaigov.net (R.S.); viroj@ihpp.thaigov.net (V.T.); thitiporn@ihpp.thaigov.net (T.S.); chompoonut@ihpp.thaigov.net (C.T.); peeraya@ihpp.thaigov.net (P.P.); 2Division of Physical Activity and Health, Department of Health, Ministry of Public Health, Nonthaburi 11000, Thailand; asawut@hotmail.com (U.A.); chonlaphan.p@anamai.mail.go.th (C.P.); 3Division of Epidemiology, Department of Disease Control, Ministry of Public Health, Nonthaburi 11000, Thailand; 4National Health Commission Office, Ministry of Public Health, Nonthaburi 11000, Thailand; weerasak@nationalhealth.or.th; 5Bureau of Health Promotion, Department of Health, Ministry of Public Health, Nonthaburi 11000, Thailand

**Keywords:** step count, step challenge, physical activity, Thailand, promotion, intervention

## Abstract

Thailand’s first national steps challenge has been implemented in 2020 with the goal to raise the level of physical activity nationwide by monitoring achievements through a smartphone application. This study examined the daily step counts of participants in the first national steps challenge. Six data points from 186,653 valid participants were retrieved and analyzed in five periods using Poisson regression. The mean daily steps peaked at 3196 in Period 1, and steadily dropped to 1205 in Period 5. The daily steps per period were analyzed using the participants’ characteristics, such as the type of participant, sex, age, body mass index, and area of residence. The overall mean daily steps of the participants meant physical activity was far below the recommended level and tended to drop in later periods. The general population achieved significantly higher mean daily steps than public health officers or village health volunteers (24.0% by multivariate analysis). Participants who were female, younger (<45 years), obese (body mass index > 30), and living in rural areas had fewer mean daily steps (13.8%, 44.3%, 12.7%, and 14.7% by multivariate analysis, respectively), with statistical significance. In the future, the national steps challenge should be continuously implemented by counting all steps throughout a day, using more strategies to draw attention and raise motivation, advocating for more participants, as well as reporting the whole day step counts instead of distance.

## 1. Introduction

The World Health Organization (WHO) describes physical activity as any bodily movements produced by skeletal muscles that requires energy expenditure [1]. Physical inactivity is the fourth leading risk factor for premature death from non-communicable diseases (NCDs) [2]. There is evidence showing that regular physical activity is associated with reduced risk of heart disease, stroke, breast and colon cancer, and diabetes, as well as improved mental health and quality of life [3,4,5]. Increasing physical activity could prevent at least 3.2 million NCD-related mortalities per year, globally [2]. However, in 2010, 23% of adults (aged 18 or above) worldwide did not meet the global recommended level of physical activity [6], which is to achieve moderate intensity physical activity for at least 150 min per week, or 75 min of vigorous intensity physical activity per week [1], equivalent to approximately 7000 steps/day [7]. In Thailand in 2016, 29% of adults had insufficient physical activity [8].

In response to the global burden from physical inactivity, the Global Action Plan on Physical Activity 2018–2030 (GAPPA) [9] was adopted by the World Health Assembly in 2018, with the goal of a 15% relative reduction (from the 2016 baseline) in the global prevalence of physical inactivity by 2030. The plan reiterates the need for a whole-of-society response to create social, cultural, and economic environments that are conducive for physical activity. The Thailand Physical Activity Strategy 2018–2030 [8] was developed through a strong participatory process and endorsed by the Cabinet in August 2018, with the strategic objectives to promote active people, create a conducive environment for active lifestyles, and develop active supporting systems. The National Steering Committee was also established to facilitate the implementation of the strategy and for effective coordination across stakeholders.

Various international experiences have shown [10,11,12,13,14,15] that using pedometers supported a moderate increase in physical activity and step-based recommendations. Kang et al.’s meta-analysis suggested a moderate to high effect of pedometer use for older adults, adults, and children (effect size = 0.53, 0.72, and 0.78, respectively), with the overall mean effect size equaling 0.68 or an average increase of 2000 steps in the intervention group for all studies [12]. An evaluation of a population-level, incentive-based intervention, promoting step counts across two Canadian provinces, concluded that a multicomponent step-based intervention increased the daily step counts (115.7 more steps compared with baseline at study week 12 and 1223.7 steps per day in physically inactive, high engagers) [13]. A study in Australia showed high engagement from participants using a smartphone application alone or in addition to the website (hazard ratio = 0.86 and 0.63 of a reduced attrition risk in web and app users and app-only users compared to web-only users) [11]. Likewise, Chaudhry et al.’s systematic review and meta-analysis illustrated that the effects of step-count monitoring interventions led to short- and long-term step-count increases from the baseline by 1126 steps/day within 4 months, and 434 steps/day within 4 years [15]. Additionally, the intervention was well received by a large segment of the population, as evidenced by the increasing number of participants in four consecutive rounds of the Singapore National Steps Challenge (129,000, 356,000, 696,000, and 810,000 progressively) [14].

Thailand’s national steps challenge policy has been initiated in 2019 by the National Steering Committee on physical activity [16] based on the concept of walking as a central component of physical activity promotion efforts [15,17,18,19,20,21,22,23]. The Thailand National Steps Challenge Season 1 was the countrywide program, implemented from 1 February to 31 March 2020. This had been initiated by the Ministry of Public Health and aimed to promote awareness of physical activity, with the goal of each participant walking or running 60 km within 60 days. Participants were asked to register and send daily distance data via their smartphones.

To date, there is no prior study that assessed the effect of a national physical activity policy at the country level in Thailand, especially not step-based and smartphone technology interventions. This article aims to analyze the magnitude of daily step counts among all participants in Season 1 and the associations between the participants’ demographics (type of participant, sex, age, body mass index, and area of residence) and step counts. It is hoped that the findings from this study will help shape the design of future national policy, promoting physical activity.

## 2. Materials and Methods

### 2.1. Study Site and Program Description

Thailand National Steps Challenge Season 1 was a countrywide program, implemented from 1 February to 31 March 2020. Participants were asked to register and send daily distance data via their smartphones to the LINE application, which is widely used in Thailand, with approximately 45 million subscribers [24,25]. The registration required information on name, sex, age, body weight, height, address, and identification card (ID card). The daily distance was tracked by the built-in smartphone accelerometer and demonstrated through health applications. Participants would send distance data to the LINE application up to 4 times per day and 45 km per report. The process of registration and distance reporting are illustrated in Figure 1 [18].

The program was primarily aimed at public health personnel and village health volunteers (VHVs) aged 18–60 years, but was also open for others who were interested in the program. A certificate was issued to participants who achieved 60 km in 60 days. Special awards, i.e., finishing T-shirts and medals, were given to the first 5000 men and 5000 women who had achieved 60 km, and to the first 1000 men and 1000 women who had achieved 100 km. The daily distance of all participants can be publicly viewed through the program website. Awards were given to ten provinces that had the most numbers of VHVs registered and participating. During the program, participants received the biofeedback on cumulative distance with a message like “well done, let’s do more steps” every time they sent the distance to the LINE application. The program was widely promoted through mass media and Ministry of Public Health’s regional, provincial, and district health facilities [17]. Although the Thailand Steps Challenge Season 1 aimed to promote physical activity, it seems that the program focused on recreational activities (such as leisure walking and running) more than overall non-specific physical activity throughout a day [18].

### 2.2. Data Sources

This study used website-based data [17], which recorded the participants’ achievement. The dataset on 31 March 2020 was the overall data of the program. We retrieved six datasets for analysis of average daily steps in five respective periods. These six datasets were recorded on (i) 21 February, (ii) 28 February, (iii) 9 March, (iv) 16 March, (v) 23 March, and (vi) 31 March. We used the dataset on 21 February as the first measurement because it was the first time that a unique identifier (bib number) was assigned to each participant. We used the bib number to link the same individual in different periods. Those five periods were different in length (7–9 days). We attempted to address the varying durations of the data collected by transforming the data in each period into a week duration; in the final analysis, we used the mean daily step change (which will be detailed in the later subsections). Each dataset contained the cumulative distance in kilometers.

Initial analysis found that approximately half of the registered participants sent no distance data to the server over the 60-day period. To reduce selection bias by including the inactive participants in the analysis, we selected only “active” participants who sent data every time in those six datasets, the “completer” as shown in Figure 2. We excluded participants aged below 18 years since children and adults had different physical activity recommendations [1] and the number of participants aged below 18 was very small (*n* = 1817, 0.004%). Participants aged above 80 years were also excluded to avoid a possibility that children had submitted data on their behalf [26,27]. Although the program system set a maximum distance of 45 km per report and allowed four entries, we set a more realistic maximum distance of <45 km/day (approximately a full marathon a day) because anything more than this was considered unreasonably high (e.g., technology bug or false report) [13,28]. Out of the total 392,565 registered participants in the server, only 186,653 participants with complete data were included in this study.

### 2.3. Data Analysis and Variable Management

Although the program used “distance” as unit of measurement, the primary outcome variable of the study was the “mean daily steps” of participants during 1 February to 31 March 2020 (using the cumulative distance from the dataset of 31 March 2020), as the mean daily steps better reflect an adequate physical activity level based on various studies [7,29,30]. The data were transformed from km to number of steps by using the mean stride distance of 116.7 cm and 126.6 cm in male and female participants aged 18–69 years, respectively [31], and at 39.6 cm and 32.7 cm in male and female participants aged 70 years or above, respectively [32], based on the following equation:Steps of male participants aged 18–69 years = (distance (km) × 1000 m × 100 cm)/116.7 cm;Steps of female participants aged 18–69 years = (distance (km) × 1000 m × 100 cm)/126.6 cm;Steps of male participants aged ≥ 70 years = (distance (km) × 1000 m × 100 cm)/39.6 cm;Steps of female participants aged ≥ 70 years = (distance (km) × 1000 m × 100 cm)/32.7 cm.(1)

The key independent variables were participant profiles: type of participant, sex, age, body mass index (BMI), and area of residence. Participants were categorized into types: (i) public health officers or VHVs (the target of the program) and (ii) general population. Age was categorized into (i) 18–45 years old and (ii) 46–80 years old (the median age was about 45). BMI was grouped into (i) non-obese (BMI < 30 kg/m^2^) and (ii) obese (BMI ≥ 30 kg/m^2^) according to the WHO definition [33]. Area of residence was categorized by postcode as (i) urban areas (Greater Bangkok and all headquarter districts) and (ii) rural areas (non-headquarter districts).

Moreover, the 6 datasets were divided into 5 periods and treated as a categorical variable: (i) Period 1 (21–27 February); (ii) Period 2 (28 February–8 March); (iii) Period 3 (9–15 March); (iv) Period 4 (16–22 March); and (v) Period 5 (23–31 March). Although the length of each period was different, we analyzed daily steps to make the steps in each period comparable. The estimated mean daily steps for each period helped refine the program on maintaining user engagement and allowed comparison of mean daily steps across these five periods. Then, we analyzed the mean daily steps per period by participant profiles (type of participant, sex, age, body mass index, and areas of residence).

Univariate analysis, using Chi-square (for categorical variables) and Student’s t-test (for continuous variable), was performed to compare the characteristics of the public health officers and VHVs with those of the general population. The overall mean daily steps by type of participants were analyzed by univariate Poisson regression. We also performed multivariate analysis to assess the effect of daily steps by accounting for the influence of all covariates by multivariate Poisson regression. The results of 186,653 participants were presented in terms of the crude incidence rate ratio (IRR), adjusted IRR, and 95% confidence interval (CI).

All analyses were performed using STATA software version 14, StataCorp, College Station, TX, USA. (serial number = 10,699,393).

### 2.4. Ethical Consideration

The dataset used in this study is from one of the national physical activity-promoting programs conducted by the Department of Health, Ministry of Public Health. As mandated by Public Health Ministerial Regulations 2009 [34], the Department of Health is required to develop surveillance system for health behavior and health impact evaluation and is not required to obtain signed consent forms from respondents. The data for this study was retrieved from a public website that cannot be mined to obtain confidential individual data, i.e., ID card numbers; thus, it was not necessary to obtain ethics approval from the Institute for the Development of Human Research Protections. The researcher followed all ethical standards in research; individual information was kept confidential and not reported in the paper.

## 3. Results

### 3.1. Baseline Characteristics

In total, we acquired 186,653 records. The majority of participants were female (about 80% of the total), as shown in Table 1. The mean age was 43.9 years (standard deviation (SD) = 11.3 years). The median age was 45.0 years (interquartile range (IQR) = 17.0 years). Non-obese participants accounted for 63.0% of all participants and over 72.3% of the participants lived in rural areas. Statistical significance was found in all characteristics when comparing between public health officers, VHVs, and the general population by Chi-square (for the categorical variables, i.e., sex, age groups, BMI groups, and area of residence) and Student’s t-test (for the continuous variable, i.e., age) with a *p* < 0.001.

### 3.2. Trends in Mean Daily Steps over Five Periods

The mean daily steps peaked at 1301 in Period 1 and steadily dropped to 633 in Period 5. This decreasing trend could be observed when comparing the participant characteristics over the five periods, as illustrated in Table 2 and Figure 3, Figure 4, Figure 5, Figure 6 and Figure 7. The average daily steps among the general population were higher than the public health officers and VHVs in each period, except in Period 4. In addition, between Periods 1 and 5, the mean daily steps for the public health officers and VHVs dropped considerably more than participants from the general population (64% and 57%, respectively). Across all periods, the average daily steps of the participants who were male, aged 46–80, non-obese, and lived in urban areas, were higher than the female participants, those aged between 18 and 45 years, obese persons, and people living in rural areas. The average daily steps in each period showed a significant difference when assessing against each characteristic by univariate Poisson with a *p* < 0.001. In addition, in females, the mean daily steps between Periods 1 and 5 dropped more than in males (64% and 55%, respectively). Participants aged between 18 and 45 years had a greater decrease in mean daily steps than the older age group (71% and 55%, respectively). A marked drop in mean daily steps was also observed in obese persons and rural dwellers, compared with their counterparts.

### 3.3. Participants’ Profile and Mean Daily Steps: Univariate and Multivariate Analysis

Based on the Poisson regression, the findings revealed that the general population had significantly higher mean daily steps (22.4%) compared with public health officers or VHVs. A similar effect could be seen in the multivariate analysis, showing that that general population had significantly higher mean daily steps (24.0%). The effect tended to be stronger when focusing on age (39.0% in the univariate analysis and 44.3% in the multivariate analysis). Furthermore, being female meant having a significantly lower mean daily steps (16.8% and 13.8% by univariate and multivariate analysis, respectively). Likewise, being obese and living in rural areas significantly decreased the overall mean daily steps (as reflected by the crude IRR and adjusted IRR being less than 1), as shown in Table 3.

## 4. Discussion

This research is among the very first studies that determined the outcomes that the national step-based intervention had on physical activity in Thailand. The study shows that the mean daily steps were far below the recommended physical activity level (approximately 7000 steps/day) and tended to drop over time. Moreover, the general population were likely to have higher overall mean daily steps than public health officers or VHVs, overall and in each period. Being female, younger, obese, or living in rural areas, were correlated with a significant decrease in overall mean daily steps, overall and in each period.

Interestingly, the mean daily steps peaked at 1301 in Period 1 and dropped steadily to 633 in Period 5 [7,30]. This might be explained by the fact that some participants might not carry their smartphone throughout the day [13] or only reported additional steps from their baseline [15]. Additionally, the decline in daily steps reflected behavioral decay of the participants. This might be due to the design of the awards that focused on competitive value more than positive reinforcement on everyday healthy behavior. For example, finishing T-shirts and medals for the first 5000 men and 5000 women who had achieved 60 km might discourage participants from continuing after achieving the program goal. Moreover, awards for provinces that had the most numbers of VHV registrations reflected a focus on registration numbers rather than ongoing participant engagement. Moreover, the COVID-19 pandemic undoubtedly disrupted domestic physical activity: the Thailand national physical activity survey in 2020 found a 55% decrease in adequate physical activity levels during March to May 2020 [35], during the “lock-down” in responses to the COVID-19 pandemic in Thailand [36]. National curfew policies were implemented, including city lockdowns, stay-at-home measures, and the closure of public places, to reduce physical gathering (recreational parks, stadiums, gyms, and other sport venues [35,37]. Thus, many participants could not reach the recommended benchmark. In the future, the program should be redesigned to include more challenging goals and promotional strategies, e.g., 10,000 steps per day, a loyalty or health points system that accumulates for tiers of rewards, and more incentives. Sub-challenge events should also be included in addition to the main individual challenges, such as monthly thematic challenges, community challenges, and corporate challenges, where participants pair up with families, friends, and colleagues [10,13,14,38], as well as physical activity at home initiatives during and after the COVID-19 pandemic [35].

Despite a marginal increase in mean daily steps, this challenge may contribute to a positive effect on health. Kang et al.’s meta-analysis study concluded that step-based interventions in all studies had increased the activity levels by approximately 2000 steps from the baseline [12]. However, even small amount of steps from only recreational activity were useful, according to a systematic review from Hall et al., suggesting that an additional 1000 steps per day can reduce all-cause mortality and the risk of cardiovascular disease [15]. Hence, in the future, national step challenge policy should take a comprehensive physical activity concept, including all steps occurring throughout the day (during household chores, occupational requirements, child care, errands, and transportation) [39]. This will also encourage persons who do not regularly engage in any sports or exercise events to increase their daily steps [30,38].

The result that the general population tended to have higher mean daily steps than public health officers or VHVs was unexpected. This might be due to the design of program, which targeted public health personnel and VHV registrations. Therefore, public health officers and VHVs might have less motivation to actually carry out the step challenge than the general population, who joined the program on a voluntary basis. In other words, the general population self-selected into the program, creating a “selection bias” [40]. Such a bias was in itself problematic, as it indirectly flagged that the program should introduce more strategies to boost active behavior among public health officers and VHVs, since they are expected to be active and healthy role models or change agents in the community.

Women had significantly lower daily steps, as shown in many national surveys on physical activity, and women also had lower physical activity levels, especially in exercise or sports [41,42,43,44]. To overcome this barrier, the program should recognise all physical activity, so that all steps throughout the day can be counted. This might better recognise women’s lifestyles and credit more steps to women [39,41,42,43].

In terms of age, being young was correlated with significantly lower daily steps. This was surprising, as previous national surveys on physical activity showed higher adequate physical activity levels among younger than older adults [41,42,43,44]. However, this study finding was in line with a meta-analysis that reported that the effect of a pedometer use was moderate to high for older adults, relative to juveniles and young adults [12]. Innovative methods to promote physical activity in young adults, such as using a built-in smartphone accelerometer should be implemented. Other strategies, such as introducing a reward system or corporate challenge, might engage the interest of various population groups [10,13,14,38]. Unsurprisingly, being obese was correlated with a smaller step count compared with non-obese persons, which might be explained by the fact that obesity impedes body movement and is associated with other chronic conditions. This finding was in line with the outcomes in prior literature [41,42]. It suggests a lack of health-promotion design that suits at-risk participants (such as obese persons and women) [38]. A step-based intervention focusing on walking and running may not be suitable for obese persons compared with other low-impact activities (such as biking and swimming) [1,45,46].

The results also showed that people who lived in rural areas had significantly lower daily steps. Although national surveys on physical activity showed less daily energy expenditure in people living in urban areas, it was clear that urban people spent more energy on recreational activities [41,42,47]. Additionally, it was the first time Thailand required people to register and send distance data to the LINE application via smart phones. It is possible that the inconvenience of using a device explains the inaccurate reporting by rural participants. Thus, traditional ways of promoting physical activity (i.e., community recreational clubs in villages or parks) are necessary, in parallel with digital and innovative campaigns [9,48].

This first Thailand national steps challenge showed many promising aspects: an innovative way to promote physical activity via smartphones, engaging a large number of participants, and gaining a small but significant number of daily steps. The Thai Ministry of Public Health should continue this steps challenge policy, with some adjustments; for instance, counting all steps throughout a day [1,46], using other strategies to attract people’s attention, and raising the motivation of various groups of participants [10,13,14,38], as well as networking with other agencies (such as the Ministry of Education, Ministry of Labor, and civic groups) to reach a wider group of people [8,16,45]. In terms of program monitoring and evaluation, “step” and “distance” can be both monitored using smartphones, non-expensive smart wrist bands, or smart watches; also, the program should use “step” as unit of measurement rather than “distance”, as it better reflects an adequate physical activity level and reduces errors from using the mean stride distance to convert distance to steps [7,29,30]. Moreover, the reporting system should upload the step data everyday with the use of the bib or registration numbers from the start of the program.

We found both strengths and limitations in this study. The strengths include the use of a countrywide database, which allowed us to assess the magnitude of the intervention outcomes via a great diversity of datasets. Secondly, this study had a large number (186,653) of participants, with valid reports across the country. Thirdly, controlling for several covariates, such as the type of participant, sex, age, BMI, and area of residence, helped minimize the bias in the estimated results, which might be due to confounders. However, the results of this population-based study should be interpreted with caution because some limitations still remain. First, the distance data used in the analysis were acquired from self-reporting, which might create a chance that the participants did not submit the data to the application every day or sent inaccurate distance data. Secondly, the equation that we used for converting distance to steps was based on the mean stride distance using male and female Korean adults [31] and male and female Chinese elderly [32]. Thus, it might not fully match Thai stride distance. Thirdly, the first available online dataset that was applicable for the analysis started from 21 February 2020 (the first date when the bib number was available). This means that the trend in daily steps before this date cannot be calculated due to a lack of a unique identifier. Fourthly, there was no baseline distance; therefore, the authors cannot determine if the intervention changed the distance or steps per day. Finally, there were other demographic data that were not included in the database, such as occupation, underlying diseases, and exact geographical residence, all of which might reveal other aspects of the data.

## 5. Conclusions

The overall mean daily steps of the participants were far below the recommended physical activity level and tended to drop in later periods. The general population had significantly more mean daily steps than the public health officers or VHVs, overall and in each period. Females, younger participants, obese persons, and rural residents had significantly lower mean daily steps than their counterparts, overall and in each period. The Ministry of Public Health should continue this steps challenge policy with some adjustments to make the policy suitable to the lifestyle of diverse participants. These include counting all steps throughout a day, motivating various groups of participants, and advocating for more participants by networking with other stakeholders. Furthermore, the program should use “step” as unit of measurement rather than “distance”, and the data should be uploaded and reported everyday with the unique identifier, made available since the first day of the program, in order to allow a trend analysis.

## Figures and Tables

**Figure 1 ijerph-17-08433-f001:**
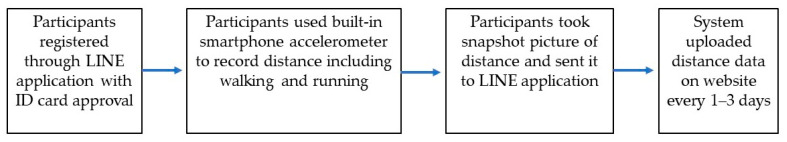
Process of registration, distance reporting, and website uploading.

**Figure 2 ijerph-17-08433-f002:**
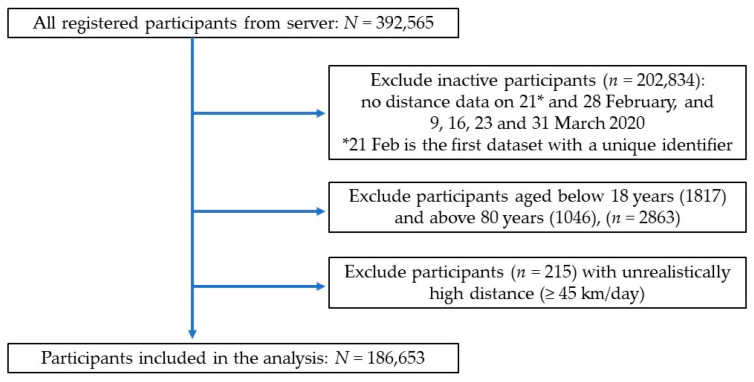
Flowchart of the data selection and analysis.

**Figure 3 ijerph-17-08433-f003:**
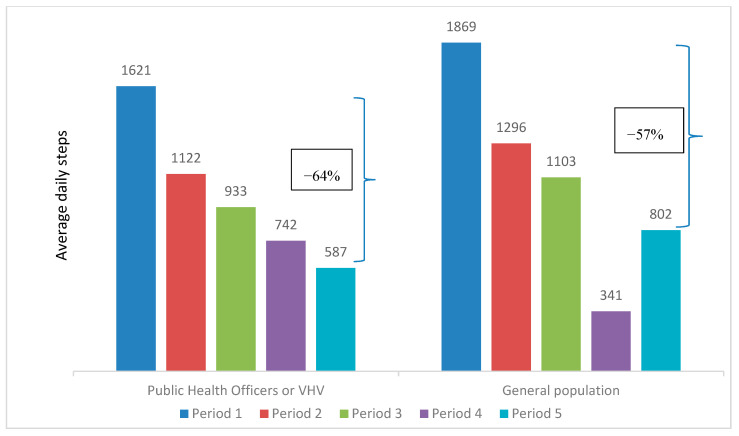
Mean daily steps by type of participant in each period. * *p* value < 0.001 for all average daily steps of public health officers or village health volunteer (VHV) and general population in each period.

**Figure 4 ijerph-17-08433-f004:**
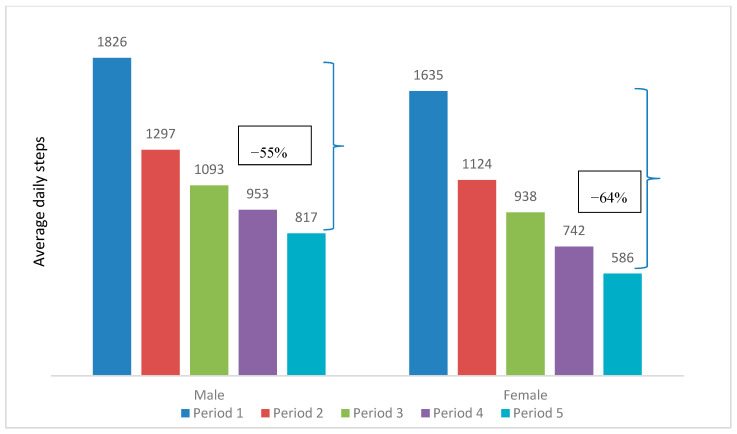
Mean daily steps by gender in each period. * *p* value < 0.001 for all average daily steps of male and female participants in each period.

**Figure 5 ijerph-17-08433-f005:**
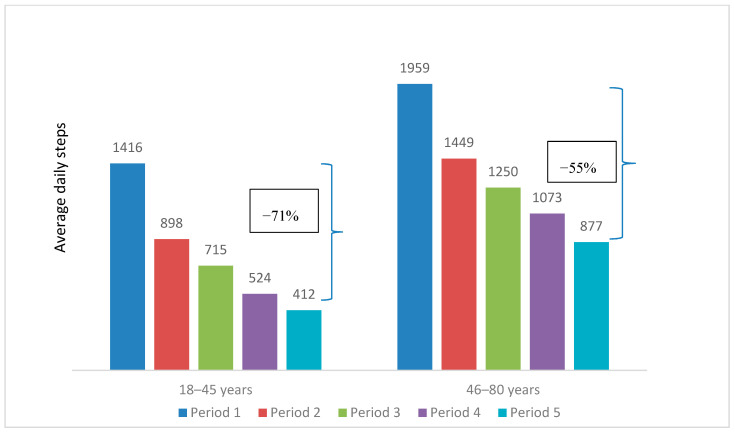
Mean daily steps by age groups in each period. * *p* value < 0.001 for all average daily steps of participants aged 18–45 and 46–80 years in each period.

**Figure 6 ijerph-17-08433-f006:**
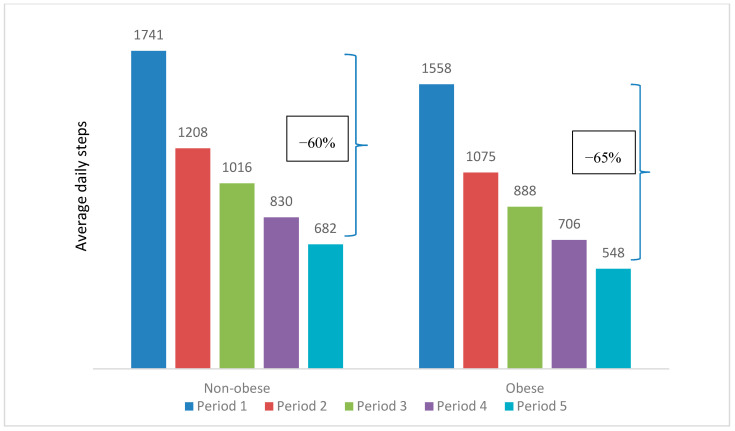
Mean daily steps by BMI groups in each period. * *p* value < 0.001 for all average daily steps of underweight or normal, overweight, and obese participants in each period.

**Figure 7 ijerph-17-08433-f007:**
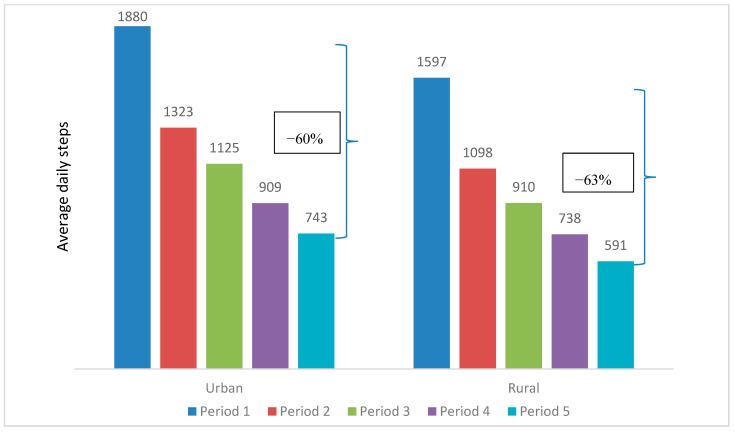
Mean daily steps by area of residence in each period. * *p* value < 0.001 for all average daily steps of participants lived in urban and rural areas in each period.

**Table 1 ijerph-17-08433-t001:** Comparing the type of participant by personal attributes.

Variables	Overall Participants (%)	Public Health Officers or Village Health Volunteers (%) *	General Population (%) *
**Sex**			
Male	37,466 (20)	23,513 (16)	13,953 (35)
Female	149,187 (80)	123,362 (84)	25,825 (65)
**Age**			
Mean (SD)	43.9 (11.3)	44.9 (11.1)	40.2 (11.5)
Median (IQR)	45 (17)	46 (17)	40 (18)
**Age groups**			
18–45	98,100 (53)	71,546 (49)	26,554 (67)
46–80	88,553 (47)	75,329 (51)	13,224 (33)
**BMI groups**			
Non-obese	117,845 (63)	91,168 (62)	26,677 (67)
Obese	68,808 (37)	55,707 (38)	13,101 (33)
**Area of residence**			
Urban	50,721 (27)	36,729 (25)	13,992 (35)
Rural	135,932 (73)	110,146 (75)	25,786 (65)

* A *p* value < 0.001 for each characteristic.

**Table 2 ijerph-17-08433-t002:** Mean daily steps overall and in each period by personal attributes.

Variables	Overall (SD) *	Period 1 (SD) *	Period 2 (SD) *	Period 3 (SD) *	Period 4 (SD) *	Period 5 (SD) *
**Overall**	1301 (1701)	1674 (2662)	1159 (2133)	969 (2084)	785 (2000)	633 (1899)
Min–Max	0–56,065	0–90,638	0–105,064	0–90,284	0–73,373	0–84,381
**Type of participants**						
Public health officers or Village Health Volunteers	1241 (1641)	1621 (2633)	1122 (2098)	933 (2047)	742 (1939)	587 (1822)
Min–Max	0–56,065	0–90,638	0–96,330	0–79,408	0–73,373	0–84,381
General population	1520 (1887)	1869 (2756)	1296 (2252)	1103 (2212)	941 (2202)	802 (2150)
Min–Max	0–50,529	0–73,135	0–105,064	0–90,284	0–97,505	0–41,758
**Sex**						
Male	1503 (1980)	1826 (2944)	1297 (2358)	1093 (2323)	953 (2364)	817 (2322)
Min–Max	0–45,721	0–90,202	0–77,957	0–79,408	0–54,488	0–84,381
Female	1250 (1619)	1635 (2585)	1124 (2072)	938 (2019)	742 (1895)	586 (1774)
Min–Max	0–56,065	0–90,638	0–105,064	0–90,284	0–73,373	0–57,576
**Age groups**						
18–45	1098 (1284)	1416 (2188)	898 (1677)	715 (1637)	524 (1486)	412 (1414)
Min-Max	0–31,046	0–35,493	0–105,064	0–38,279	0–36,645	0–37,001
46–80	1525 (2043)	1959 (3078)	1449 (2514)	1250 (2458)	1073 (2414)	877 (2295)
Min-Max	0–56,065	0–90,638	0–38,424	0–90,284	0–73,373	0–84,381
**BMI groups**						
Non-obese	1361 (1769)	1741 (2720))	1208 (2201)	1016 (2166)	830 (2101)	682 (2017)
Min–Max	0–56,065	0–90,202	0–105,064	0–90,284	0–73,373	0–84,381
Obese	1198 (1570)	1558 (2555)	1075 (2009)	888 (1935)	706 (1811)	548 (1672)
Min–Max	0–40,622	0–90,638	0–52,000	0–45,059	0–43,687	0–35,191
**Area of residence**						
Urban	1485 (1815)	1880 (2721)	1323 (2297)	1125 (2258)	909 (2159)	743 (2073)
Min–Max	0–50,111	0–73,135	0–105,064	0–90,284	0–73,373	0–50,107
Rural	1232 (1650)	1597 (2635)	1098 (2066)	910 (2013)	738 (1935)	591 (1828)
Min–Max	0–56,065	0–90,638	0–77,957	0–79,408	0–67,505	0–84,381

* A *p* value < 0.001 for all the mean daily steps of each personal attributes overall and in each period.

**Table 3 ijerph-17-08433-t003:** Daily mean steps: univariate and multivariate analysis.

Variables	Univariate Analysis		Multivariate Analysis	
	Crude Incidence Rate Ratio	95% Confidence Interval *	Adjusted Incidence Rate Ratio	95% Confidence Interval *
**Type of Participants**				
● General population (ref = public health officers or VHVs)	1.2243	1.2239–1.2247	1.2399	1.2395–1.2402
**Sex**				
● Female (ref = male)	0.8316	0.8313–0.8318	0.8617	0.8614–0.8620
**Age group**				
● 46–80 years (ref = 18–45 years)	1.3895	1.3892–1.3899	1.4433	1.4430–1.4434
**BMI group**				
● Obese (ref = non-obese)	0.8804	0.8802–0.8806	0.8726	0.8723–0.8728
**Area of residence**				
● Rural (ref = urban)	0.8293	0.8291–0.8295	0.8529	0.8527–0.8531

* A *p* value < 0.001 for all variables.

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
