# Peer review of "Daily Step Counts from the First Thailand National Steps Challenge in 2020: A Cross-Sectional Study"

_ijerph, 2020, doi:10.3390/ijerph17228433_

Round 1

Reviewer 1 Report

Thank you for the opportunity to review the manuscript which evaluates a national step challenge in Thailand. The paper offers a interesting overview of the program and can contribute to the gaps in the literature. However, there several issues that should be addressed: 

Major Concerns: 

  1. Page 2 Lines 63-85. Introduction. The description of the intervention belongs in the methods section of the manuscript instead of the introduction. A more in depth review of the literature is needed in the introduction (expand lines 57-62). 
  2. Page 3 Line 109-115. Methods. Please provide justification for exclusions for those >80yo and those with implausible values. Are these based on existing literature or any other evidence? Please provide reference if possible. 
  3. Page 4 Line 119-121. Methods. Of major concern is how the primary outcome variable is expressed (i.e., distance translated to step count using average stride length). This method does not account for differences in stride length for individuals. Hence comparing demographic groups that are known to have differ stride lengths is a significant issue. For example, females, older adults, and obese adult can all plausibly have shorter stride lengths due to differences in physical stature and biomechanics. Do these results hold if distance is examined as the dependent variable? 
  4. Page 5 Line 155. This sentence is unclear. Please clarify if statistical test are comparing within subgroups or across groups (i.e. PHO/VHV vs. general population). 
  5. Page 5 Line 164-176. Throughout this paragraph the authors sate the 'average daily steps were higher/lower for one group compared to another'. It is unclear if these are statistically significant - what is the p-value for these comparisons? Also, the authors must clarify if they are referring to the overall average or the average at each period. If referring to period, several of these statements are inaccurate. 
  6. Regarding baseline, does a true baseline measure exist for the sample? I appears that the first measurement point is weeks after the intervention started. If so, these is a limitation that warrants exploration. 
  7. Page 10 Line 281-299. Limitations. The exclusion of individuals with missing data is not a strength of the study. It is exclusion criteria (i.e., only include those with complete data). Those excluded from the analysis are likely difference from those that completed. Was this examined?  And using the stride length of a middle-aged male is a significant limitations that warrants additional attention; especially since it is not representative of the study's sample. 

Minor Concerns: 

  1. Page 4 Line 119. Should this be '21 Feb - 31 March' instead of 1 Feb? 
  2. Page 4 Lines 129-132. Please elaborate on the selected time periods. Why do the differ in length (i.e. 7 vs 9 days)? How could this impact the findings? Why were 5 periods selected? This is unclear in the manuscript. 
  3. Page 5 Line 155. Village Health Volunteer - change to VHV and be consistent throughout the manuscript. 
  4. Page 5 Line 158. Please provide a justification for why underweight and normal weight participants were combined.
  5. Page 6 Line 173-175. This sentence is unclear and appears to be incorrect based on figure 5. 
  6. Page 9 Line 225-229. The authors note that the step challenge should account for comprehensive physical activity. It is not clear in their methods section that the step challenge includes all PA or just exercise. Please elaborate. 

Author Response

27 October 2020

Dear  reviewer,

Many thanks for the comments and suggestions; we found them very useful and constructive in shaping the revision of our manuscript. We had responded point by point to all comments into the revised manuscript. Please find our responses below.

Best wishes,

Thitikorn Topothai

Corresponding author on behalf of all authors.

Reviewer 2 Report

The current manuscript provides an analysis of data accumulated by the national program – “Thailand National Steps Challenge Season 1”.  The analysis presents a few issues with usefulness in its current state, but the data itself is well worth publication.

Major Revisions:

  1. The results present a large novelty effect from the ‘Thailand National Steps Challenge Season 1’. Please perform relevant statistics (comparing period 1 data to period 5 – currently this is just a magnitude description) and discuss this in the conclusions.
  2. Was IRR calculated using the entire population’s data or just registered participant’s data? This is necessary information to describe in the methods.
  3. Additional descriptive statistics should be performed such as standard deviations or confidence intervals for average daily steps during each period.

Minor Revisions:

  1. The title should read “Daily Step Counts from the First Thailand National 3 Steps Challenge in 2020: A Cross-Sectional Study
  2. The Introduction paragraphs that describe the ‘Thailand Steps Challenge Season 1’ should be moved to the methods in 2.1.
  3. The reason for separating by ‘General Population’ versus ‘Public Health Officers or VHV’ should be explained.
  4. While the manuscript is well organized, there are a many reoccurring English grammar errors that requires additional copy editing.

Subjective Revisions

  1. Utilizing the variable ‘age’ as categorical data instead of continuous could introduce Simpson’s Paradox.
  2. I believe the conclusions to be biased in favor of the program despite falling short of guideline recommendations and the likely novelty effect. A less biased stance is okay to take while maintaining the belief that the program is useful for improving at least a portion of the population’s physical activity.

Author Response

(The authors gave the same response as above.)

Round 2

Reviewer 1 Report

Thank you for the opportunity to review the revised manuscript which evaluates a national step challenge in Thailand. The paper offers a interesting overview of the program and can contribute to the gaps in the literature. I believe the authors have responded adequately to the prior comments. However, some pending comments remain:

Line 126: Revise sentence for clarity. 

Please refrain from using the term 'baseline' when referring to data collected during phase 1 of the intervention. It is not baseline but the first measurements taking during the intervention. There is no true baseline measurement. Therefore it can not be determined if the intervention changed distanced traveled or steps per day. This must be clearly stated. 

With respect to the dependent variable (mean daily steps), the authors need to provide justification for translating the distanced traveled in kilometers as measured via accelerometry in phone. In addition to Table 2, it would be helpful to examine the if distanced traveled was significantly different across groups observed.

Also, it would be very helpful to see the range in mean daily step counts. The steps reported are low in comparison to other studies (Althoff et al. 2017, Nature). 

Author Response

3 November 2020

Dear editor and reviewer, IJERPH

Many thanks for the second-round comments and suggestions by the reviewer; we found them very useful and constructive in shaping the revision of our manuscript. We had responded point by point to all comments into the revised manuscript. Please find our responses below.

Best wishes,

Thitikorn Topothai

Corresponding author on behalf of all authors.
